# Ligand requirements for immunoreceptor triggering
Michael I. Barton[1], Rachel L. Paterson [1,2], Eleanor M. Denham[1,3], Jesse Goyette [1,4,5] &
Philip Anton van der Merwe [1,5] ✉

Leukocytes interact with other cells using cell surface receptors. The largest group of such receptors are non-catalytic tyrosine phosphorylated receptors (NTRs), also called immunoreceptors. NTR signalling requires phosphorylation of cytoplasmic tyrosine residues by SRC-family tyrosine kinases. How ligand binding to NTRs induces this phosphorylation, also called NTR triggering, remains controversial, with roles suggested for size-based segregation, clustering, and mechanical force. Here we exploit a recently developed cell-surface generic ligand system to explore the ligand requirements for NTR triggering. We examine the effect of varying the ligand's length, mobility and valency on the activation of representative members of four NTR families: SIRPβ1, Siglec 14, NKp44 and TREM-1. Increasing the ligand length impairs activation via NTRs, despite enhancing cell-cell conjugation, while varying ligand mobility has little effect on either conjugation or activation. Increasing the valency of the ligand, while enhancing cell-cell conjugation, does not enhance activation at equivalent levels of conjugation. These findings are more consistent with a role for size-based segregation, rather than mechanical force or clustering, in NTR triggering, suggesting a role for the kinetic-segregation model.

The cell surface of an immune cell or leucocyte presents many different receptors, which sense their environment through ligand binding[1–3]. Many leucocyte receptors bind to ligands on the surface of other cells to mediate adhesion and/or transduce signals which regulate leucocyte function. These signals determine whether the leucocyte ignores or responds to the cell and influence the nature of the response. The largest class of such receptors are non-catalytic tyrosine-phosphorylated receptors (NTRs), which are also called immunoreceptors[4]. More than one hundred leucocyte receptors, in more than 20 families, can be classified as NTRs[4]. Because they regulate immune cell function, NTRs have roles in a wide range of diseases, and they are being exploited for therapeutic purposes. For example, synthetic NTRs (e.g. chimeric antigen receptors) and antibodies targeting NTRs or their ligands (e.g. checkpoint inhibitors) have become standard therapies for several forms of cancer[5,6].

While NTRs have structurally diverse extracellular regions they all have, or are associated with signalling subunits that have, conserved tyrosine containing motifs in their cytoplasmic domains, such as the immunoreceptor tyrosine-based activation motif (ITAM) and immunoreceptor tyrosine-based switch motif (ITSM)[4]. These motifs are phosphorylated by SRC-family tyrosine kinases, which are tethered via acyl groups to the inner leaflet of the plasma membrane. This phosphorylation is regulated by the receptor tyrosine phosphatases CD45 and CD148, which act on both SRC-kinases and their substrates. While the extracellular regions of NTRs are structurally diverse, they are typically smaller (4-10 nm) than many abundant cell surface molecules such as CD43, CD44, CD45, CD148 and integrins, which range in size from 21-50 nm. When these size differences were first noted for the TCR and its peptide-MHC (pMHC) ligand it was predicted that, when T cells contacted other cells, there would be segregation of the TCR/pMHC complex from larger molecules like CD45[1]. This observation, together with evidence that constitutive tyrosine phosphatase activity suppresses TCR triggering in resting cells[7,8], led to the proposal that TCR binding to pMHC induced tyrosine phosphorylation of the TCR by trapping it in small regions of close contact which exclude large receptor tyrosine phosphatases CD45 and CD148 but not the SRC-kinases[9]. This mechanism was subsequently termed the kinetic segregation (KS) model[10]. Subsequent studies from multiple laboratories using a wide range of techniques have demonstrated that the KS mechanism plays a key role in TCR triggering[11–21]. Recently it has been shown that synthetic receptors based on the TCR, namely chimeric antigen receptors (CARs), also appear to trigger by the KS mechanism, which has important implications for the design of these receptors and selection of their target antigens[22]. Other mechanisms that been proposed to contribute to TCR triggering are aggregation[23] or

[1]Sir William Dunn School of Pathology, University of Oxford, Oxford, UK. [2]Present address: Stemmatters, Biotecnologia e Medicina Regenerativa SA, Parque de Ciência e Tecnologia Avepark, Zona Industrial da Gandra, Barco, Portugal. [3]Present address: Enara Bio, The Magdalen Centre, Oxford Science Park, 1 Robert Robinson Avenue, Oxford, UK. [4]Present address: Department of Molecular Medicine, School of Biomedical Sciences, University of New South Wales, Sydney, NSW, Australia. [5]These authors contributed equally: Jesse Goyette, Philip Anton van der Merwe. ✉e-mail: anton.vandermerwe@path.ox.ac.uk

conformational change, with conformational change being either allosteric[24] or induced by mechanical force[25,26].

The similarities in signalling between the TCR and other NTRs have led to the hypothesis that the KS mechanism may contribute to triggering by other NTRs[4,27]. Indeed, evidence for this has been reported for NKG2D in NK cells[28], Dectin-1[29] and FcγRs[30] in macrophages, and CD28 in T cells[31]. However, the diversity of NTRs and their ligands, and the fact that many NTR ligands have yet to be identified, has hampered investigation of the triggering mechanism in a wider range of NTRs. We have recently developed a generic ligand system based on the SpyTag/SpyCatcher split protein, which enables cell-surface Streptactin to be used to engage any NTR incorporating a membrane-distal StrepTagII peptide[32]. Importantly, this generic ligand stimulated TCRs at the same surface density as the native TCR ligand, validating it as a suitable model system for investigating NTR triggering[32].

In the present study we used this generic ligand system to explore the ligand requirement for triggering by representative members of 4 distinct NTR families (Fig. 1A): Signal regulatory protein β1 (SIRPβ1), Sialic acid-binding immunoglobulin-type lectin 14 (Siglec 14), Natural killer receptor 44 (NKp44) and Triggering receptor expressed on myeloid cells 1 (TREM-1). SIRPβ1 is highly homologous to the inhibitory receptor and therapeutic target SIRPα (Kharitonenkov et al.[33]). SIRPβ1 and SIRPα are examples of paired activatory/inhibitory NTRs[4], with conserved extracellular regions but distinct transmembrane and cytoplasmic domains. The native ligand of SIRPβ1 is unknown but it is thought to promote phagocytosis in macrophages (Hayashi et al.[34]). Siglec 14 is a member of the sialic acid-binding Siglec family of receptors[35,36]. Siglec 14 and Siglec 5 are paired activatory and inhibitory NTRs, respectively (Angata et al.[36]). NKp44 is important for the activation and cytotoxic activity of natural killer cells and reportedly binds a wide range of ligands[37]. Finally, TREM-1 is an activatory member of a family of NTRs[38] that has been reported to bind peptidoglycan recognition protein 1 (PGLYRP1) (Read et al.[39]) and extracellular actin (Fu et al.[40]). All four receptors associate with, and presumably signal using, the ITAM-containing adaptor protein DAP12 (Lanier and Bakker[41]).

To investigate the triggering mechanism used by these NTRs we examined the effect of changing ligand size, mobility and valency on activation. In addition to using the same generic ligand, we used the same cell type and readouts for all NTRs, to facilitate comparison. Elongating the ligand inhibited activation via these receptors despite enhancing receptor/ligand mediated cell-cell conjugation. In contrast, changing the ligand mobility had little effect on conjugation or activation. Finally, while increasing the ligand valency increased cell-cell conjugation as well as activation, it decreased the level of activation at equivalent levels of cell-cell conjugation. Taken together, these results are more consistent with a role for the KS mechanism in triggering by these NTRs, and do not support a role for mechanical force or clustering.

## Results

### Varying ligand length
In the SpyTag-SpyCatcher system a covalent (isopeptide) bond spontaneously forms between Spytag and SpyCatcher when they are mixed together (Zakeri et al.[42]). In our previously described generic ligand system[32], SpyTag is fused to the N-terminus of a transmembrane protein expressed on ligand-presenting CHO cells, forming the ligand anchor, while SpyCatcher is fused to a Strep-Tactin tetramer, which can have from one to four active binding sites, as required (Fig. 1A). The receptor is modified at its membrane-distal N-terminus to contain a Strep-tag II peptide, which binds monovalent Strep-Tactin with a $K_D$ of 43 μM[32]. This is within the affinity range typical of leucocyte cell-cell interactions[3,43].

To test the effect of ligand length on activation via NTRs, we produced CHO cells expressing either a short or a long ligand anchor, where the latter includes a ~ 11 nm spacer comprising the four Ig domains of CD4 (Fig. 1A)[44]. By titrating the monovalent Strep-Tactin SpyCatcher we produced a panel of CHO cells with a range of binding sites. We used a previously described method to accurately quantify the number of Strep-

Tactin binding sites presented by these cells[32]. This involved measuring the maximum number of biotin binding sites and the $K_D$ for Strep-Tactin SpyCatcher coupling to cells (sFig. 1) and using these parameters to calculate the number of binding sites, as described in the "Materials and methods" section.

We first measured the impact of ligand elongation on the receptor engagement using a cell-cell conjugation assay as a readout. For receptor cells we used THP-1 cells expressing the SIRPβ1 receptor with an N-terminal Strep-tag II peptide. THP-1 cells are a human monocytic cell line widely used as a model for studying monocyte/macrophage function[45]. They are a suitable model as SIRPβ, TREM-1, Siglec-14 and NKp44 are all known to be expressed on monocytes. The conjugation assay involves mixing receptor and ligand cells stained with different fluorescent dyes and measuring double positive events by flow cytometry (Fig. 1B, upper right quadrants). As expected, reducing the number of binding sites resulted in a decrease in the percentage of receptor cells in conjugates (Fig. 1B).

We then compared the conjugation efficacy of SIRPβ1 THP-1 cells mixed with short or long CHO cells presenting different numbers of binding sites. When plotting conjugates against binding sites, cells presenting the long ligand produced more conjugates (Fig. 1C, left panel). This suggests that increasing the ligand length promotes conjugate formation, which is consistent with other studies suggesting that increasing the length of short (< 8 nm) cell surface ligands improves receptor engagement[11,31,46].

We next examined the effect of ligand length on SIRPβ1 mediated interleukin 8 (IL-8) production. IL-8 is inflammatory chemokine produced monocytes in response to a wide variety of activation signals[47]. It is widely used a convenient and reliable readout of THP-1 activation, with low levels of basal release and a rapid increase following stimulation[48]. Interestingly the long ligand stimulated less IL-8 production than the short ligand (Fig. 1C, middle panel), despite mediating improved conjugate formation. To normalise for differences in conjugate formation we plotted the functional IL-8 response against the percentage of receptor cells in conjugates (Fig. 1C, right panel). This confirmed that, at equivalent levels of conjugate formation, the long ligand induced lower levels of IL-8 production.

We performed conjugation and stimulation assays on other NTRs (Siglec 14, NKp44 and TREM-1) using the short or long generic ligand (Fig. 2). As in the case of SIRPβ1, the results for NKp44 and TREM-1 show an increase in conjugation efficacy when binding to the long ligand compared to the short ligand (Fig. 2, left data panels). This was not the case for Siglec 14, where no difference was seen (Fig. 2B, left data panel). Elongation of the ligand impaired activation of IL-8 release via all four NTRs, both before and after normalising for conjugate formation (Fig. 2, centre and right data panels, respectively).

One likely explanation for the effect of ligand length on activation efficiency is that increasing the ligand length increases the length of the NTR/ligand complex. If this is the case, activation should also be reduced by increasing the length of the NTRs. The extracellular regions of SIRPβ1 and Siglec 14 each contain 3 immunoglobulin superfamily (IgSF) domains, whereas the extracellular regions of NKp44 and TREM-1 only have 1 IgSF domain (Fig. 1A). To examine whether this size difference had a measurable impact we reanalysed the data in Fig. 2 to enable comparison of conjugation and activation via NTRs exposed to the same ligand (sFig. 3). Short NTRs (NKp44 and TREM-1) mediated lower levels of conjugation (sFig. 3, left panels) but high levels of stimulation when normalised for conjugation (sFig. 3, right panels), and this was observed with both short (sFig. 3A) and long (sFig. 3B) ligands. Taken together, these data show that elongation of these NTR/ligand complexes attentuates activation via these four NTRs.

### Varying ligand mobility
We next examined the effect of varying the ligand anchor on NTR activation. We compared ligand anchors based on the transmembrane and cytoplasmic domains of CD80, CD43 and the glycosylphosphatidylinositol (GPI) anchor of CD52. The CD43 cytoplasmic domain interacts with the actin cytoskeleton though Ezrin/Radixin/Moesin (ERM) proteins (Yonemura et al.[49]), and so is likely to be more firmly anchored. In contrast, CD52

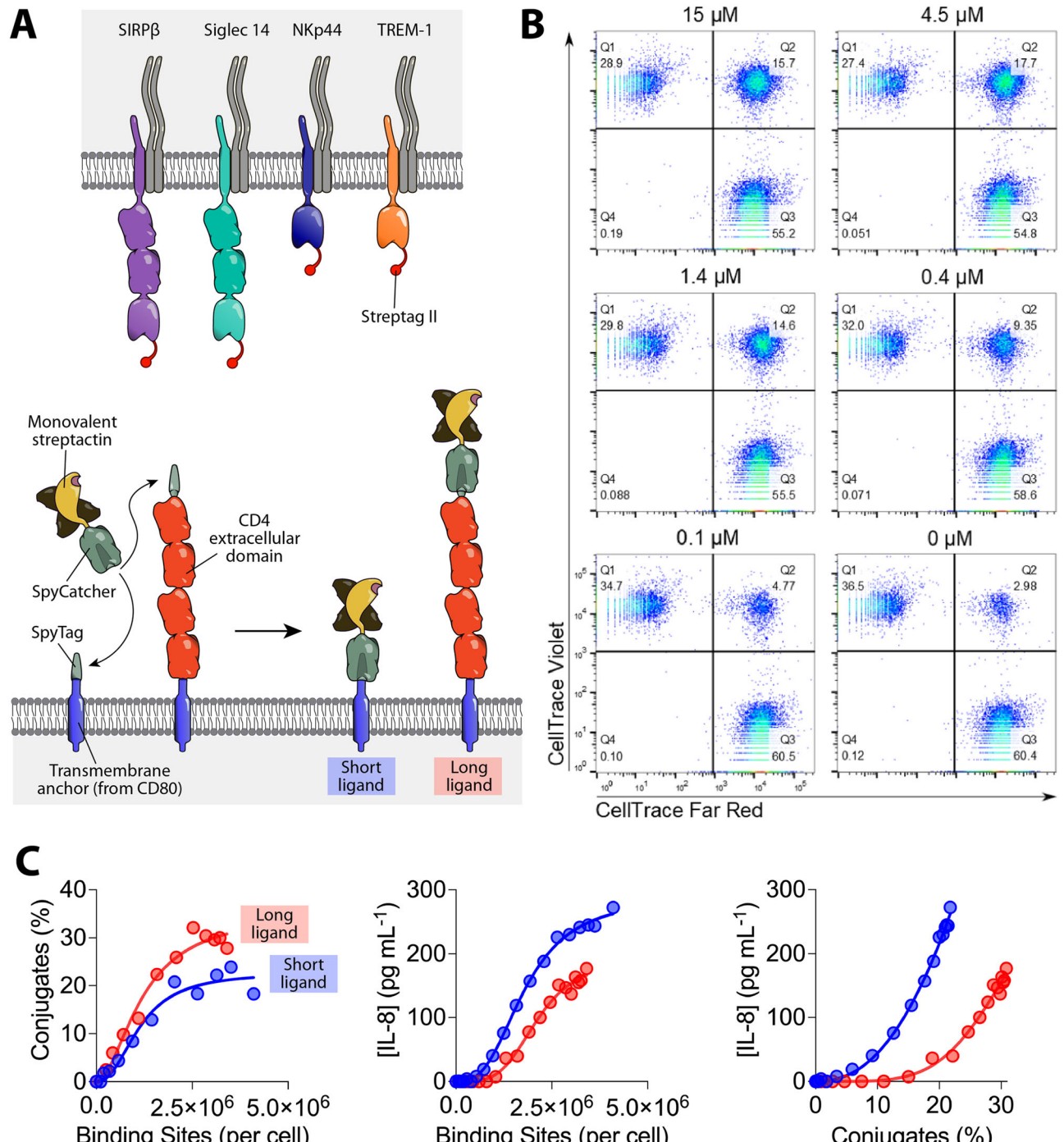

**Fig. 1 | Testing the effect of increasing ligand length on NTR activation.**
**A** Schematic depiction of the components of the generic ligand system used to test the effect of increasing ligand length and valency. The shared DAP-12 signalling homodimer is grey. With tetravalent streptactin (not shown) all four of the binding sites are able to bind Steptag II. **B** Representative flow cytometry data from a conjugation assay between SIRPβ1 expressing THP1 receptor cells and CHO cells expressing short ligands coupled with the indicated concentration of monovalent Strep-Tactin SpyCatcher. Receptor and ligand cells were stained with CellTrace Violet and Far Red, respectively, and events in the upper right quadrant were presumed to be conjugates. **C** Conjugation of (left panel) and IL-8 secretion by (middle

panel) SIRPβ1 expressing THP1 cells incubated with CHO cells expressing the indicated number of short (blue) or long (red) monovalent ligands, measured as described in the "Materials and methods" section using parameters determined in sFig. 1. IL-8 secretion is plotted against conjugation in the right panel. These are representative results from three independent experiments, which are combined in Fig. 2A for statistical analysis. Due to experimental constraints the stimulation and conjugation assays in this (and subsequent) experiment(s) were performed on successive days. The same result was obtained when performed on the same day (sFig. 2).

is a GPI anchored protein and thus less firmly anchored, and presumably more mobile (Fig. 3A). To examine mobility, SpyCatcher-GFP was coupled to ligand anchors and fluorescence recovery after photobleaching (FRAP) performed (sFig. 4A). As expected, the CD52 anchor conferred greater

mobility than the CD80 or CD43 anchor, which were similar (sFig. 4A and Fig. 3B).

We next compared the effect of changing the ligand mobility on conjugation with and stimulation of SIRPβ1 expressing THP-1 cells. It

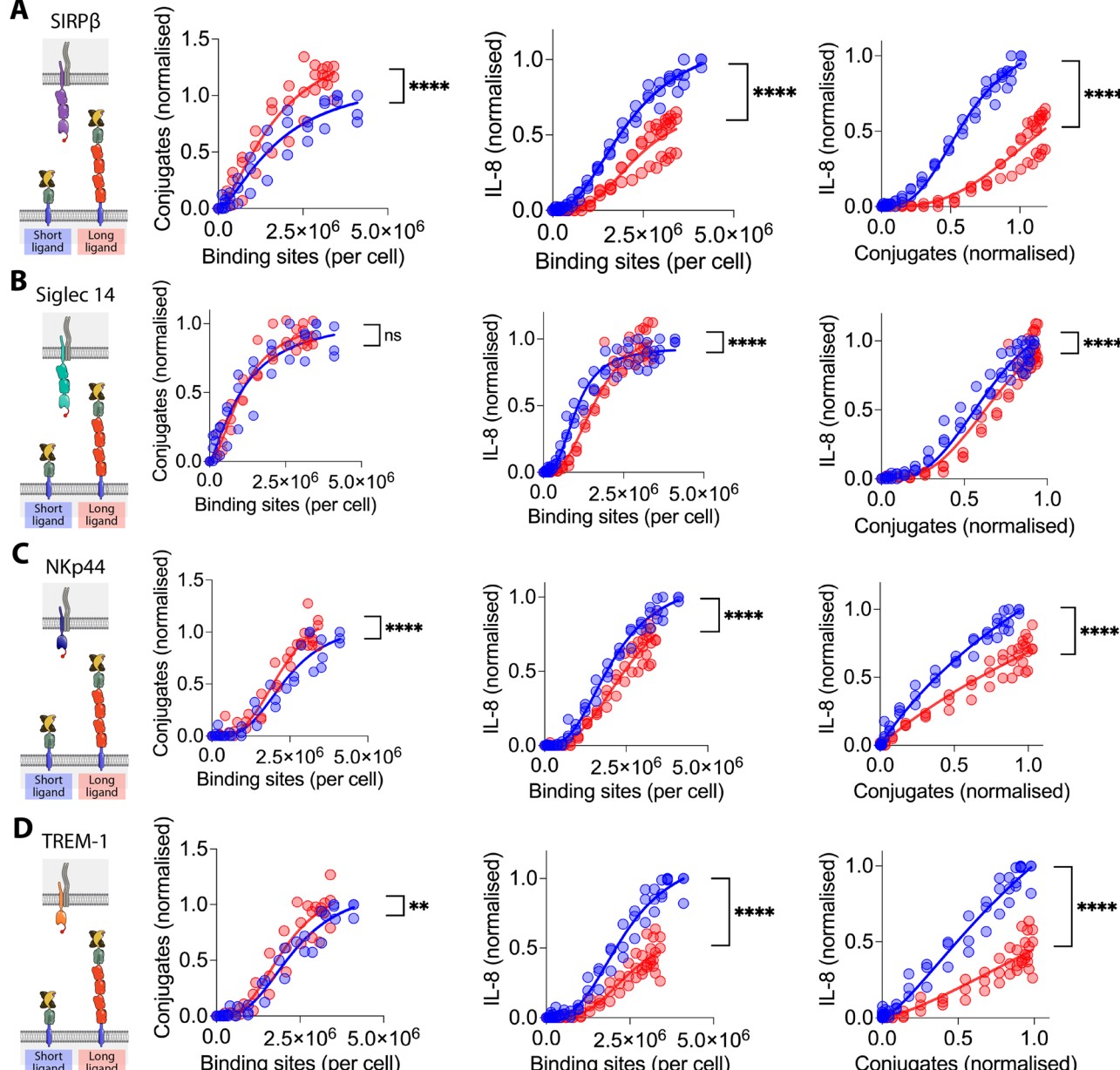

**Fig. 2 | The effect of NTR ligand length on activation via four NTRs.** THP-1 cells expressing (**A**) SIRPβ1, (**B**) Siglec 14, (**C**) NKp44 or (**D**) TREM-1 with N-terminal StrepTagII peptides were incubated with CHO cells expressing the indicated numbers of short (blue) or long (red) monovalent generic ligand binding sites and conjugate formation (left data panel) and IL-8 release (middle data panel) measured. Ligand binding sites were determined as described in the Materials and Methods using parameters determined in sFig. 1. The IL-8 release versus conjugation level is plotted in the right panels. The data from three biological replicates (including one SIRPβ1 replicate shown in Fig. 1) are plotted with the data normalised to the level of conjugation or stimulation achieved with the short ligand within each replicate. The data were fitted as described in the "Materials and methods" section and an F test was used to test the significance of differences between the fits.

was not possible to attain as high a level of binding sites on the CD43 and CD52 ligand anchor cells but comparison was possible over a reasonable range. CD52 and CD43 anchored ligands induced similar levels of conjugate formation (Fig. 3C, top panel) and IL-8 release (Fig. 3C, middle panel) at comparable levels of binding sites. CD52 anchored ligand induced slightly less IL-8 release than CD43 anchored ligands when plotted against levels of conjugation (Fig. 3C, bottom panel). However this difference was small and could be the result of faster turnover of the CD52 anchored ligand (sFig. 5), which would reduce engagement during the 20 h stimulation. While the CD80 anchored ligand was less potent at mediating conjugation and IL-8 release, this was not a consequence of differences in lateral mobility which was similar to the lateral mobility of the CD43 anchored ligand (Fig. 3B). Taken

together, these results suggest that changes in the ligand mobility do not affect SIRPβ1 mediated conjugation or triggering.

**Varying ligand valency**

Since ligand-induced clustering of receptors is often assumed to be the mechanism of receptor-activation, we next examined the effect of increasing the valency of the ligand from 1 to 4 by using a tetravalent form of Strep-Tactin Spycatcher. As expected, conjugation with tetravalent instead of monovalent Strep-Tactin Spycatcher resulted in a four-fold increase in the number of binding sites (sFig. 6). We then compared the ability of mono-valent and tetravalent ligand to mediate conjugation and induce IL-8 secretion from THP-1 cells expressing the 4 different NTRs. Tetravalent ligand induced conjugation via all 4 NTRs at lower ligand binding site

numbers than monovalent ligand (Fig. 4, left data panels), indicating that increasing the ligand valency increases NTR binding, presumably by increasing avidity. Increasing the valency enabled activation of all four receptors, as measured by IL-8 release, at much lower ligand binding sites (Fig. 4, middle data panels). However, after controlling for increased conjugate formation, tetravalent ligand was actually less effective than monovalent ligand at stimulating IL-8 release at equivalent levels of conjugation (Fig. 4, right panels). The same result was observed with the CD80, CD43 and CD52 ligand anchors (sFig. 7 and 8). These results indicate that, while increasing the valency of a cell surface-associated ligand enhances binding to NTRs, it does not increase activation via NTRs.

We next investigated whether activation of NTR by the high avidity tetravalent Strep-Tactin SpyCatcher was sensitive to ligand length. THP-1 cells expressing four representative NTRs were exposed to CHO cells presenting tetravalent Strep-Tactin SpyCatcher on either short or long CD80 anchors (Fig. 5). The short ligands were less effective at mediating conjugation than the long ligand for two (SIRPβ1 & NKp44) of the four NTRs (Fig. 5, left panels), but more effective at stimulating IL-8 production for three of the four NTRs (SIRPβ1, NKp44 and TREM-1), both before (Fig. 5, centre panels) and after (Fig. 5, right panels) controlling for conjugate formation. These results show that even high avidity NTR/ligand interactions remain sensitive to ligand length for three of the four NTRs studied here.

## Discussion

We have exploited our previously described generic cell surface ligand system[32] to explore the effects of varying ligand length, mobility, and valency on activation of four NTRs, SIRPβ1, Siglec-14, NKp44 and TREM-1. One advantage of this system is that it enables titration of ligand surface density, enabling detection of quantitative differences in the ability of cell-surface ligands to mediate conjugation and stimulation. A second advantage is that it enables multiple NTRs to be assessed using the same set of ligands, increasing throughput and facilitating comparisons between NTRs. Comparison was also enabled by expressing them in the same cell type and using the same functional readout. A third advantage is that it enables analysis of orphan NTRs, such as SIRPβ1, whose ligands have yet to be identified.

Our first key finding is that elongation of generic ligands abrogated activation of all four NTRs. This was not a consequence of decreased binding as elongated ligands mediate enhanced cell-cell conjugation. While this contrasted with results in a supported lipid bilayer (SLB) system, in which elongation of CD48 abrogated CD2 binding[50], it is consistent with results obtained with cell surface expressed ligands, including CD48[11,28,31,46]. A likely explanation for this is that ligands on cell surfaces, unlike ligands on SLBs, are crowded by the larger molecules present at high densities. Our finding that long NTRs were less effectively activated than short NTRs suggest that the increased NTR/ligand length abrogates NTR signalling. These data are most consistent with the KS mechanism of NTR triggering[27], since increasing the NTR/ligand length would be expected to increase the inter-membrane distance and thus reduce segregation of inhibitory receptor tyrosine phosphatases such as CD45 from the engaged NTR[11]. Numerous studies have confirmed that increasing receptor/ligand length abrogates CD45 segregation from engaged NTRs[11,22,30,31,51,52]. In one of these studies an elongated high affinity TCR ligand, derived from the OKT3 monoclonal antibody, was able to activate TCRs despite less efficient exclusion of CD45[51]. However, no titration of the ligand number was performed, and a lower affinity variant of the same ligand was unable to activate T cells[51].

An alternative explanation for our finding that elongation abrogates activation through NTRs is that this could decrease the level of force experienced by the NTR upon ligand engagement[53]. These data are therefore also consistent with models of NTR triggering postulating that a mechanical force imposed upon ligand binding alters the conformation of the NTR[25,26]. However, whether changing ligand length affects the force experienced by an NTR has yet to be confirmed, and it remains unclear how such a conformational change of NTRs could be transmitted through the membrane to enhance phosphorylation of their cytoplasmic domains. Such a mechanism is difficult to reconcile with enormous structural variability of NTRs[4] and the

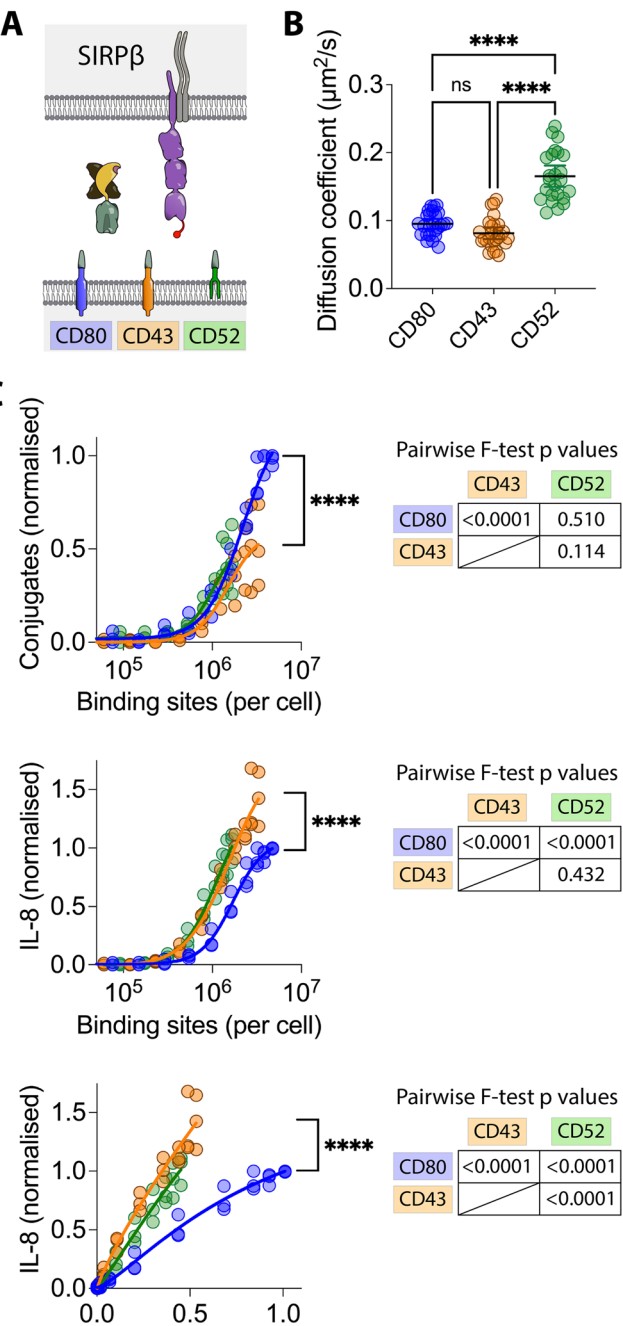

**Fig. 3 | Effect of ligand anchor on SIRPβ1 stimulation. A** Schematic depiction of the components of the generic ligand system used to test the effect of varying the ligand anchor. **B** The diffusion coefficients of the different ligand anchors were measured by FRAP after coupling GFP-Spycatcher. The mean and SD from three independent experiments were compared by ANOVA. **C** THP-1 cells expressing SIRPβ1 with an N-terminal StrepTagII peptide were incubated with CHO cells expressing the indicated number of ligand binding sites with CD80 (blue), CD43 (orange) or CD52 (green) anchors, and conjugate formation (top panel) and IL-8 release (middle panel) measured. Ligand binding sites were determined as described in the "Materials and methods" section using parameters determined in sFig. 4. The IL-8 release versus conjugation level is plotted in the bottom panel. The data from three biological replicates are plotted with the data normalised to the level of conjugation or stimulation achieved with the CD52 ligand anchor within each replicate. These data were fitted as described in the "Materials and methods" section and an F test was used to test the significance of differences between the fits collectively and pairwise (Tables).

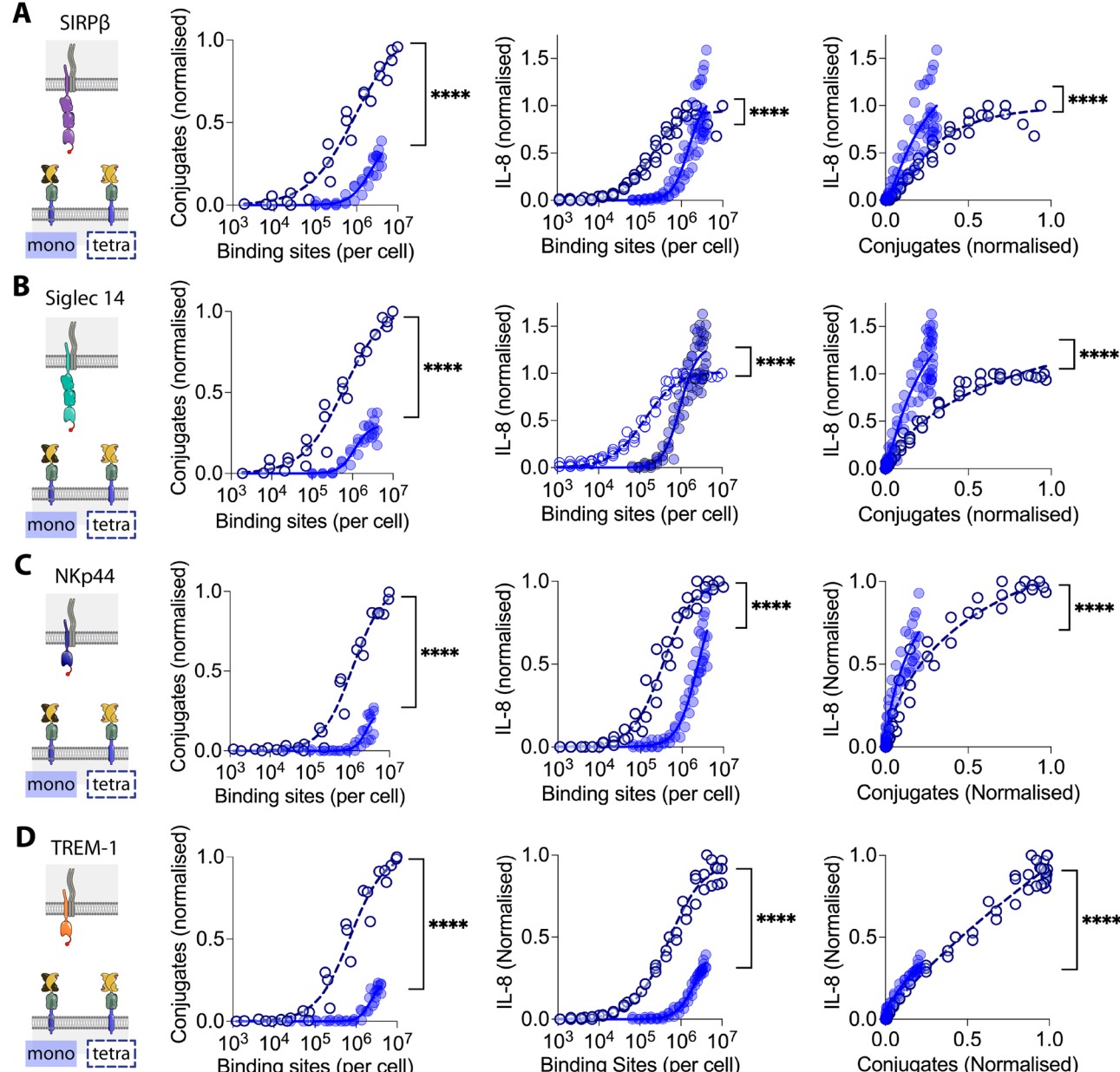

**Fig. 4 | Effect of ligand valency on NTR stimulation.** THP-1 cells expressing (**A**) SIRPβ1, (**B**) Siglec 14, (**C**) NKp44 or (**D**) TREM-1 with N-terminal StrepTagII peptides were incubated with CHO cells expressing the indicated numbers of monovalent (filled circles) or tetravalent (open circles) short generic ligand binding sites and conjugate formation (left data panel) and IL-8 release (middle data panel) measured. Ligand binding sites were determined as described in the Materials and Methods using parameters determined in sFig. 6. The IL-8 release versus conjugation level is plotted in the right panels. The data from three biological replicates are plotted with the data normalised to the level of conjugation or stimulation achieved with the short tetravalent ligand within each replicate. These data were fitted as described in the "Materials and methods" section and an F test was used to test the significance of differences between the fits.

fact that chimeric NTRs such as CARs tolerate extensive variation in the regions (hinge, transmembrane and cytoplasmic domains) that couple their ligand binding domains with their tyrosine-containing signalling motifs[54,55].

A second key finding is that changing the mobility of the ligand anchor had little impact on its ability to mediate activation via NTRs. The CD52 anchor comprises a lipid (GPI) which we show confers greater lateral mobility. While there is only a ~ 2 fold change in mobility, it should be noted that membrane diffusion in cells is far slower than in model membranes, likely because it is impaired by membrane proteins that associate with the cortical actin cytoskeleton, which have been termed 'picket fences'[56,57]. A lipid anchor also allows a ligand to be more easily extracted from the plasma membrane by force. It follows that the amount of force that can be exerted on an NTR, both tangential and perpendicular to the membrane, should be

lower with lipid-anchored than a transmembrane-anchored ligand, such as CD43, able to bind to the actin cytoskeleton through ERM proteins. Thus, our finding that CD43 and CD52 anchors were similarly effective at activating an NTR does not support a role for mechanical force in triggering for these NTRs. We note, however, that we did not directly measured force. Much stronger evidence against a role for force was reported by Göhring et al. [58], who showed that changing the lateral mobility of the TCR ligand in an SLB system by ~1000 fold had no effect on TCR triggering despite substantially changing the force experienced by the TCR[58].

The third key finding is that increasing the valency of the NTR ligand did not increase activation of the NTR after controlling for enhanced NTR binding, as assessed by cell-cell conjugation. This result contrasts with the findings obtained with soluble NTR ligands such as cross-linked antibodies

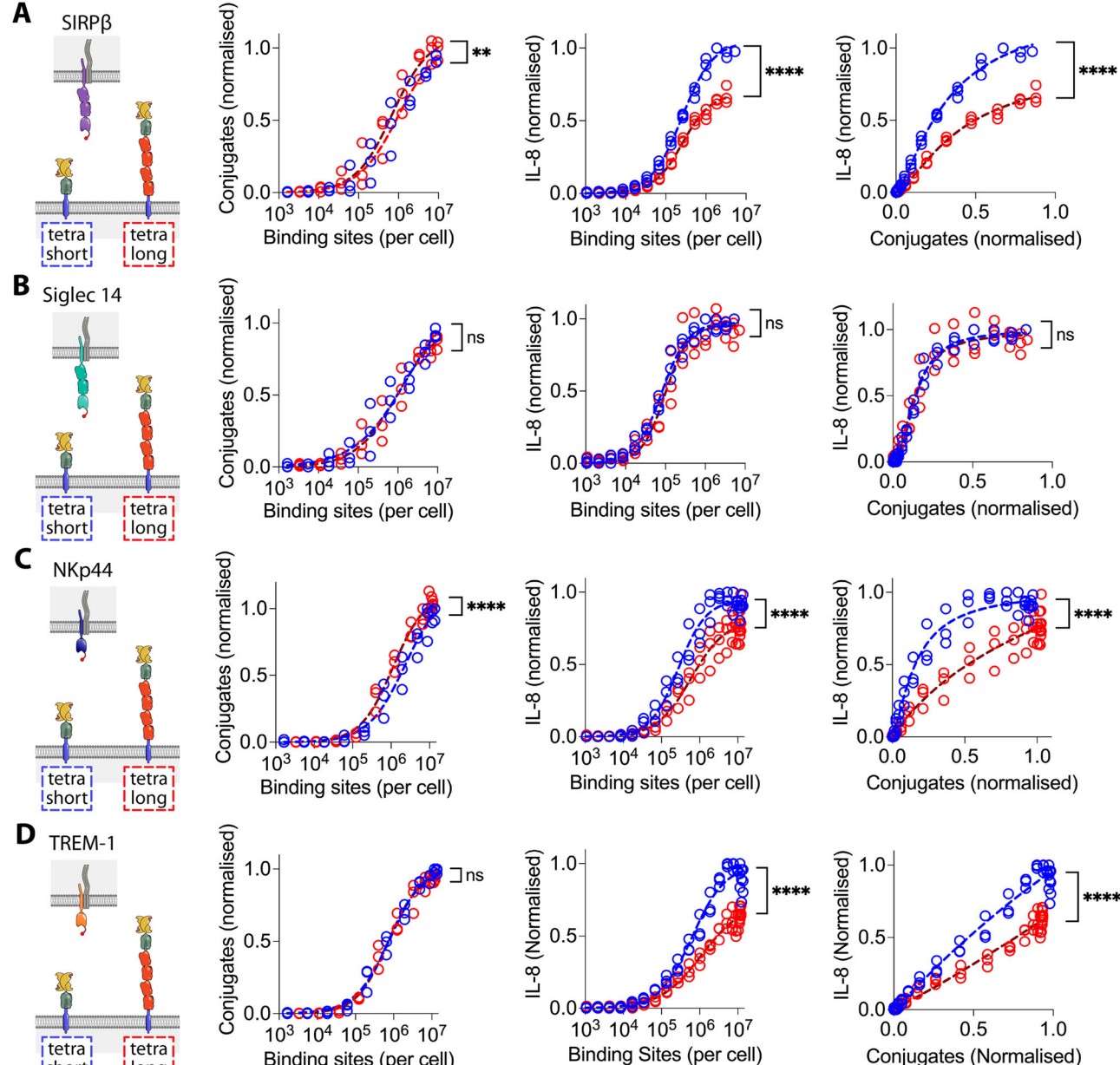

**Fig. 5 | The effect of NTR ligand length on activation by tetravalent ligands.** THP-1 cells expressing (**A**) SIRPβ1, (**B**) Siglec 14, (**C**) NKp44 or (**D**) TREM-1 with N-terminal StrepTagII peptides were incubated with CHO cells expressing the indicated numbers of short (blue) or long (red) tetravalent ligand binding sites and conjugate formation (left panel) and IL-8 release (middle panel) measured. Ligand binding sites were determined as described in the Materials and Methods using

parameters determined in sFig. 9. The IL-8 release versus conjugation level is plotted in the right panels. The data from three biological replicates are plotted with the data normalised to the level of conjugation or stimulation achieved with the short tetravalent ligand within each replicate. These data were fitted as described in the "Materials and methods" section and an F test was used to test the significance of differences between the fits.

and natural ligands engineered to be multivalent, where increasing the valency is required for NTR triggering[59,60]. One possible explanation for this discrepancy between the effect of valency with soluble and surface associated ligand is that a soluble multivalent ligand could, by forming clusters of NTRs, exclude molecules with bulky ectodomain, such as CD45, from clustered NTRs, whereas soluble monovalent ligands are unable to do this. In contrast, even a monovalent surface-associated ligand can trap the NTR in zones of a close intermembrane contact from which CD45 and CD148 are excluded. While our finding that tetravalent ligands are less effective at activating NTRs than monovalent ligands at equivalent levels of NTR binding requires confirmation and further analysis, the fact that increased valency does not enhance NTR triggering argues against aggregation as the primary mechanism of physiological NTR triggering. Further evidence

against this model is that almost all cell surface NTR ligands that have been identified to date are monovalent[4]. In contrast, cell surface receptors known to signal by binding induced multimerisation, such as class III tyrosine kinase receptors[61] and TNF receptor superfamily members[62], typically have multivalent ligands.

Our finding that elongation of a tetravalent generic ligand did not impair activation via Siglec 14, while it did impair activation via the three other NTRs, suggests that, for some NTRs, increasing ligand valency can bypass the need for the KS mechanism. It is noteworthy that some Siglec family members, including the Siglec-14 paired receptor Siglec-5[63], are able to form disulfide linked dimers. Dimeric Siglec-14 would allow formation of large 'zipper-like' aggregates with tetravalent generic ligand, enabling exclusion of CD45 and CD148 without needing the KS mechanism.

Taken together, our finding that activation via NTRs is abrogated by ligand elongation and aggregation and unaffected by the mobility of the ligand anchor supports our hypothesis that NTRs signal by the KS mechanism. While our results focus only on four representative NTRs, these results are likely to apply to a larger number of NTRs, including other members of the Siglec and TREM families and NTRs from other families that signal via the DAP-12 adaptor[64]. As reviewed in the Introduction, there is already evidence that many NTRs that do not associate with DAP12, including the TCR, NKG2D, CD28, and FcγR, exploit the KS mechanism for triggering. Taken together with the results presented in this paper, this suggests that the KS mechanism is used by NTRs which signal through intrinsic tyrosine motifs (CD28, FcγRI and FcγRII) and a variety of signalling subunits (DAP12, DAP10, CD3δεγ, CD247, and FcRγ).

One limitation of the present work is that we used artificial ligands rather than native ligands. While we have previously validated the generic ligand system for the T cell receptor[32], this is more challenging for other NTRs, as their ligands cannot easily be titrated. Our recent development of Combicells[65], which exploit the SpyCatcher/SpyTag system to present native ligand on antigen-presenting cells, should enable our key results to be confirmed with native ligands, where known. A second limitation, which is a consequence of the system used, is lack of imaging data of the interface between THP-1 cells and CHO cells. Advanced microscopy, beyond the scope of this study, is likely needed to image the molecular events in the microvilli-like structures and close contact areas involved in NTR triggering at cell-cell interfaces[18,66,67]. A third limitation is that we only test one prediction of the KS model, that elongation of the receptor/ligand complex will abrogate NTR triggering. Other predictions of the KS model should be tested in future work. A final limitation is that we did not examine early signalling steps, such as tyrosine phosphorylation of DAP-12. Such studies are very difficult to perform in the high-throughtput manner required for our titration-based analysis, and are unlikely to provide further insights into the the ligand requirements for activation of these NTRs.

## Methods
### Constructs
Receptor sequences for SIRPβ1, Siglec 14, NKp44 and TREM-1 were inserted into the pHR-SIN-BX-Strep-tag II plasmid as described by (Denham et al., [32]). The same constructs containing the DAP12 adaptor protein was also used.

For the ligand anchors DNA encoding the Igk leader sequence (bold), HA tag (italics), SpyTag (underlined) bracketed by GGS linkers, and the hinge, transmembrane and intracellular regions of mouse CD80, mouse CD43, or human CD52 (italics underlined) was inserted into the vector pEE14. To express the long ligand anchor DNA encoding human CD4 (italics bold) was inserted between the SpyTag and CD80 hinge. This included an R to D point mutation (underlined) to prevent CD4 binding MHC class II. After expression CD52 anchor is cleaved and linked to GPI anchor at the serine residue marked in bold.

**CD80 anchor (short).** M E T D T L L L W V L L L W V P G S T G D *Y P Y D V P D Y A T* G G S <u>A H I V M V D A Y K P T K</u> G G S G G S *H V S E D F T W E K P P E D P P D S K N T L V L F G A G F G A V I T V V V I V V I I K C F C K H R S C F R R N E A S R E T N N S L T F G P E E A L A E Q T V F L*

**CD80 anchor (long).** M E T D T L L L W V L L L W V P G S T G D *Y P Y D V P D Y A T* G G S <u>A H I V M V D A Y K P T K</u> G G S G G S ***K V V L G K K G D T V E L T C T A S Q K K S I Q F H W K N S N Q I K I L G N Q G S F L T K G P S K L N<u>D</u> D A D S R R S L W D Q G N F P L I I K N L K I E D S D T Y I C E V E D Q K E E V Q L L V F G L T A N S D T H L L Q G Q S L T L T L E S P P G S S P S V Q C R S P R G K N I Q G G K T L S V S Q L E L Q D S G T W T C T V L Q N Q K K V E F K I D I V V L A F Q K A S S I V Y K K E G E Q V E F S F P L A F T V E K L T G S G E L W W Q A E R A S S S K S W I T F D L K N K E V S V K R V T Q D P K L Q M G K K L P L H L T L P Q A L P Q Y A G S G N L T L A L E A K T G K L H Q E V N L V V M R A T Q L Q K N L***

*T C E V W G P T S P K L M L S L K L E N K E A K V S K R E K A V W V L N P E A G M W Q C L L S D S G Q V L L E S N I K V L P T R S H V S E D F T W E K P P E D P P D S K N T L V L F G A G F G A V I T V V V I V V I I K C F C K H R S C F R R N E A S R E T N N S L T F G P E E A L A E Q T V F L*

**CD43 anchor.** M E T D T L L L W V L L L W V P G S T G D *Y P Y D V P D Y A T* G G S <u>A H I V M V D A Y K P T K</u> G G S G G S *Q E S S G M L L V P M L I A L V V* <u>*V L A L V A L L L L W R Q R Q K R R T G A L T L S G G G K R N G V V D A W A G P A R V P D E E A T T T S G A G G N K G S E V L E T E G S G Q R P T L T T F F S R R K S R Q G S L V L E E L K P G S G P N L K G E E E P L V G S E D E A V E T P T S D G P Q A K D E A A P Q S L*</u>

**CD52 anchor.** M E T D T L L L W V L L L W V P G S T G D *Y P Y D V P D Y A T* G G S A H I V M V D A Y K P T K G G S G G S <u>*S D T S Q T S S P S A S S N I S G G I F L F F V A N A I I H L F C F S*</u>

**Strep-Tactin-SpyCatcher sequence.** Strep-Tactin is underlined, SpyCatcher is in italics and the polyaspartate sequence is in bold.

<u>M A E A G I T G T W Y N Q L G S T F I V T A G A D G A L T G T Y V T A R G N A E S R Y V L T G R Y D S A P A T D G S G T A L G W T V A W K N N Y R N A H S A T T W S G Q Y V G G A E A R I N T Q W L L T S G T T E A N A W K S T L V G H D T F T K V K P S A A S</u> **D D D G D D D G D D D D** *S A T H I K F S K R D E D G K E L A G A T M E L R D S S G K T I S T W I S D G Q V K D F Y L Y P G K Y T F V E T A A P D G Y E V A T A I T F T V N E Q G Q V T V N G K A T K G D A H I*

**Strep-Tactin sequence.** M A E A G I T G T W Y N Q L G S T F I V T A G A D G A L T G T Y V T A R G N A E S R Y V L T G R Y D S A P A T D G S G T A L G W T V A W K N N Y R N A H S A T T W S G Q Y V G G A E A R I N T Q W L L T S G T T E A N A W K S T L V G H D T F T K V K P S A A S

**Dead Streptavidin sequence.** Bold amino acids mark substitutions in order to prevent binding to Strep tag II or biotin.

M A E A G I T G T W Y **A** Q L G **D** T F I V T A G A D G A L T G T Y E **A** A V G A E S R Y V L T G R Y D S A P A T D G S G T A L G W T V A W K N N Y R N A H S A T T W S G Q Y V G G A E A R I N T Q W L L T S G T T E A N A W K S T L V G H D T F T K V K P S A A S

The GFP-SpyCatcher construct was described in (Denham et al., [32]).

### THP-1 cell lines
THP-1 cells (ATCC #TIB 202) were maintained in RPMI-1640 media (Sigma-Aldrich #R8758) supplemented with 10% foetal bovine serum (FBS) and 1 in 100 penicillin/streptomycin (Thermo Fisher Scientific #15140122) at 37 °C in a 5% $CO_2$ containing incubator.

### CHO cell lines
Chinese Hamster Ovary (CHO) mock cells were maintained in DMEM (Sigma-Aldrich #D6429) supplemented with 5% FBS and 1 in 100 penicillin/streptomycin. CHO ligand anchor cells (short and long) were maintained in L-Glutamine-free DMEM (Sigma-Aldrich #D6546) supplemented with 10% dialysed FBS (dialysed three times against 10 L PBS), 1 in 100 penicillin/streptomycin, 1x GSEM supplement (Sigma-Aldrich #G9785) and 50 μM L-Methionine sulfoximine (Sigma-Aldrich #M5379).

### Lentiviral transduction of THP-1 cells
Either receptor-expressing lentivector alone, or with the DAP12 adaptor-expressing lentivector, was co-transfected with the lentiviral packaging plasmids pRSV-Rev (Addgene plasmd #12253), pMDLg/pRRE (Addgene plasmd #12251) and pMD2.g (Addgene plasmd #12259) into HEK293T cells using X-tremeGENETM HP (Sigma-Aldrich 6366546001) as per the manufacturer's instructions. Two days after transfection, viral supernatants were harvested, filtered (0.45 μM syringe filter) and used for the transduction of THP-1 cells in the presence of 5 μg mL$^{-1}$ Polybrene.

## Analysing receptor and adaptor expression using flow cytometry and cell sorting by fluorescence-activated cell sorting

Cells were analysed for receptor surface expression by flow cytometry using anti-Strep-tag II antibody StrepMAB™ directly conjugated to Oyster 645 (IBA Lifesciences #2-1555-050), or unconjugated StrepMAB™ (IBA Lifesciences #2-1507-001) with anti-mouse IgG1 antibody Alexa Fluor 647 (Thermo Fisher Scientific #A-21240), on a Cytek DxP8. Introduced adaptor expression was tested via expression of EmGFP encoded on the adaptor lentivector. Cells were sorted for matched high expression of receptor plus introduced adaptor by fluorescence-activated cell sorting (FACS) (MoFlo Astrios, Beckman Coulter).

## Transfection of CHO cells with various ligand anchors

CHO cells were transfected with either pEE14 (CHO mock) or pEE14-ligand anchor (CHO ligand anchor) using Xtreme-GENE 9™ as per the manufacturer's instructions. Transfected lines were cultured in the appropriate selection media after 48 h.

## Checking ligand anchor expression by flow cytometry

CHO Cells were analysed for ligand anchor surface expression by flow cytometry using anti-HA-Tag antibody Alexa Fluor 647 (6E2; Cell Signalling Technology). Alternatively, cells were coupled with saturating concentration of monovalent StrepTactin SpyCatcher. Biotin ATTO 647 was then added at 2 μM for 30 min. The cells were washed 3 times in PBS 1%BSA before they were fixed in PBS 1% formaldehyde and analysed via flow cytometry.

## Expression and purification of monovalent and tetravalent Strep-Tactin-SpyCatcher

Strep-Tactin SpyCatcher and dead streptavidin (monovalent) or Strep-Tactin SpyCatcher and Strep-Tactin (tetravalent) expressed in Escherichia coli BL21-CodonPlus (DE3)-RIPL cells (Agilent Technologies #230280) were combined and refolded from inclusion bodies. Inclusion bodies were washed in BugBuster (Merck Millipore #70921) supplemented with lysozyme, protease inhibitors, DNase I and magnesium sulfate as per the manufacturers' instructions. Subunits were then mixed at a 3:1 molar ratio to improve the yield of the desired tetramer. The subunits were refolded by rapid dilution in cold PBS and contaminates removed via precipitation using ammonium sulfate before additional ammonium sulfate was added to precipitate the desired tetramer. Precipitated protein was resuspended in 20 mM Tris pH 8.0, filtered (0.22 μm syringe filter), and loaded onto a Mono Q HR 5/5 column (GE Healthcare Life Sciences). Desired tetramers were eluted using a linear gradient of 0-0.5 M NaCl in 20 mM Tris pH 8.0, concentrated, and buffer exchanged into 20 mM MES, 140 mM NaCl pH 6.0 (Denham et al., [32]).

## Coupling CHO generic ligand cells

Ligand anchor expressing or mock transduced CHO cells were incubated with various concentrations of monovalent or tetravalent Strep-Tactin SpyCatcher in 20 mM MES, 140 mM NaCl, pH 6.0 and 1% BSA for 10 min at RT. Unbound Strep-Tactin SpyCatcher was removed by washing three times with PBS 1% BSA.

## FACS sorting of ligand anchor expressing CHO cells

CHO short and long cells were coupled with a saturating concentration of monovalent Strep-Tactin SpyCatcher. Biotin ATTO 647 (ATTO-TEC #AD 647-71) was then added at 2 μM for one hour and the excess washed off with PBS 1% BSA. The short and long CHO cells were then sorted for matched expression of atto 647 signal corresponding to the expression level of ligand anchor using FACS (MoFlo Astrios, Beckman Coulter).

## Biotin-4-fluorescein quenching assay

The valency of purified monovalent and tetravalent Strep-Tactin Spy-Catcher was measured using biotin-4-fluorescein (Sigma-Aldrich #B9431-5MG) which when bound to Strep-Tactin become quenched. Monovalent

and tetravalent Strep-Tactin SpyCatcher was incubated with a titration of biotin-4-fluorescein in black, opaque plates for 30 min at RT in PBS 1% BSA. Fluorescence was measured ($\lambda_{ex}$ 485 nm, $\lambda_{ex}$ 520 nm) using a plate reader. Fluorescence values were corrected for background fluorescence before analysis. Negative control (buffer alone) data were fitted with the linear regression. Sample data was fitted with a segmental linear regression, equation below, where X is the biotin-4-fluorescein concentration, Y is fluorescence (AU), X0 is the biotin-4-fluorescein concentration at which the line segments intersect. Slope1 was constrained to zero and is the gradient of the first line segment, slope2 is the gradient of the second line segment. Intercept1 was constrained to zero and is the Y value at which the first line segment intersects the Y axis. Slope2 was constrained to the gradient given by the linear regression of the negative control.

$$Y1 = Intercept1 + Slope1 * X$$

$$Y \, at \, X0 = Slope1 * X0 + Intercept1$$

$$Y2 = Y \, at \, X0 + Slope2 * (X - X0)$$

$$Y = IF \, (X < X0, \, Y1, \, Y2)$$

The X0 value was converted into an estimate of the number of biotin-binding sites per tetramer using the concentration of Strep-Tactin Spy-Catcher added and assuming complete binding of biotin-4-fluorescein.

## Measurement of ligand binding sites on cells

Ligand-anchor expressing or mock transduced CHO cells ($3 \times 10^6$) were pre-incubated with a saturating concentration of monovalent or tetravalent Strep-Tactin SpyCatcher. The above biotin-4-fluorescein quenching assay was performed in the same manner but with a known number of cells. The X0 term (calculated from the curve fit using equations above) was converted to the average number of binding sites per cell using the equation below, where N is the average number of ligands per cell, X0 is the saturation concentration of biotin-4-fluorescein extracted (M), V is the sample volume (L), $N_A$ is Avogadro's constant and C is the number of cells in the sample.

$$N = \frac{X0 * V * N_A}{C}$$

To measure relative levels of coupled SpyCatcher per cell, ligand cells were pre-incubated with a range of concentrations of monovalent or tetravalent Strep-Tactin SpyCatcher or buffer alone (as a negative control) before being incubated with 2 μM biotin ATTO 488 (ATTO-TEC #AD 488-71) pre-mixed with 40 μM biotin for 30 min at RT. The presence of biotin minimises the self-quenching activity of ATTO dye observed with tetravalent Strep-Tactin. Cells were analysed by flow cytometry and the gMFI when cells were incubated with buffer alone instead of Strep-Tactin SpyCatcher was subtracted from all corresponding sample gMFI values. These values were then fitted with the single site binding model, equation below, where Y is the gMFI (AU), Bmax is the maximum specific SpyCatcher binding indicated by gMFI in AU, X is the [Streptactin-SpyCatcher] added (M) and $K_D$ is the [Spy-Catcher] that yields 50% maximal binding to CHO cells (M).

$$Y = \frac{B \max * X}{Kd + X}$$

To convert Y values into the average number of binding sites per ligand cell, the number of binding sites per cell at saturating monovalent/tetravalent Strep-Tactin SpyCatcher concentration calculated in the biotin-4-fluorescein assay was substituted into the equation above as Bmax. Y values were then re-calculated following this adjustment. This method was

followed in each independent experiment and then an average value for the $K_D$ and Bmax was used when combining replicates.

## Fluorescent recovery after photobleaching

CHO cells ($1 \times 10^5$) expressing the three ligand anchors (CD80, CD43 and CD52) were transferred to a 35 mm glass bottom dish one day before imaging. To prepare cells for imaging each dish was washed 3 times with coupling buffer plus 10% FBS. SpyCatcher-GFP was then added in excess (approximately 10 μM) and left for 10 min. Before washing the cells three times in PBS 1% BSA 10% FBS. Cells were transferred to the Olympus FV1200 laser scanning microscope with 37 °C chamber for equilibration. The 60X magnification lens was used to locate cells spread over the glass coverslip. A small area of the cell (approximately 10%) was selected a few control images were taken before a 20 s bleach performed. A time lapse series of images was then taken to track the recovery of the GFP signal.

Time lapse image series were imported into ImageJ for analysis. For each time lapse the bleach area was selected (bleach) along with a control area which was taken to be the rest of the cell contact with the glass (control area) and a negative control area around the outside of the cell (negative). Firstly, the intensity from the negative area was subtracted from the bleach and control area. The bleach area was then divided by the control area for each time frame and these values were normalised to the control image before the bleach was performed. These values were then plotted against time and the equation below used to find the half time for each ligand anchor.

$$Y = Y0 + (Plateau - Y0) * (1 - e^{-K*x})$$

Where Y0 is the value when X (time) is zero, Plateau is the Y value at infinite times and K is the rate constant from which the half time is derived.

The half time value could then be converted into the diffusion coefficient using the equation below (Soumpasis, 1983).

$$Diffusion\ coefficient = 0.224 * r^2 / half\ time$$

Where 0.224 is a constant for a circular bleach area, r is the radius of the bleach area, and the half time is the derived from fitting the one phase association equation to the bleach recovery above.

## Ligand turnover

CHO ligand anchor cells were incubated with 15 μM monovalent Strep-Tactin SpyCatcher (or buffer alone) as described above and incubated at 37 °C to match stimulation assays for the time points indicated. Cells were analysed for generic ligand surface expression using ATTO 488 biotin as above and normalised to the geometric fluorescence intensity value at time 0. To calculate the decay, the gMFI values were fitted with the equation below where Y0 is the Y value when X = 0, Plateau is the Y value at which the curve reaches a plateau, X is time in minutes, and K is the rate constant in inverse minutes.

$$Y = (Y0 - Plateau).e^{-K.X} + Plateau$$

## IL-8 production

CHO ligand anchor cells ($2 \times 10^5$) coupled with either monovalent/tetravalent Strep-Tactin SpyCatcher or buffer alone were mixed with single Strep-tag II tagged receptor and adaptor expressing THP-1 or untransduced THP-1 cells ($1 \times 10^5$) in DMEM 5% FBS, 1 in 100 penicillin/streptomycin, 2 μg mL$^{-1}$ avidin. Cells were incubated in a 37 °C 10% CO2 containing incubator for 20 h. Supernatants were harvested and assayed for IL-8 by ELISA following manufacturer's instructions (Thermo Fisher Scientific #88808688).

## Conjugation assays

Ligand cells were stained with CellTrace Far Red (Thermo Fisher Scientific #C34564) at a final concentration of 1 μM in PBS at a density of $1 \times 10^6$ cells per ml. THP-1 receptor cells were stained with CellTrace Violet (Thermo Fisher Scientific #C34557) at a final concentration of 1 μM in PBS for 20 min. CHO Ligand cells ($4 \times 10^5$) coupled with either monovalent/tetravalent Strep-Tactin SpyCatcher were mixed with ($2 \times 10^5$) THP-1 cells in PBS 1% BSA on Ice for 1 h. Conjugation efficacy was analysed by flow cytometry.

Due to experimental constraints stimulation and conjugation assays were completed on successive days. However, we confirmed that the same result was observed when the stimulation and conjugation assays were performed in parallel on the same day by splitting the cells in half for each assay (Supplementary Fig. 2).

## Data analysis, statistics and reproducibility

For receptor stimulation assays, IL-8 concentrations in negative controls (where CHO cells were pre-incubated with buffer alone instead of Strep-Tactin SpyCatcher) were subtracted from corresponding sample IL-8 concentrations to correct for background levels. Dose-response curves were then fitted with the below equation where Y is the measured cell response (pg mL$^{-1}$), Bottom and Top are the minimum and maximum cell response respectively (pg mL$^{-1}$), EC50 is the number of binding sites per cell that yields a half maximal response, X is the number of binding sites per cell and Hill slope relates to the steepness of the curve.

$$Y = Bottom + \frac{(X^{Hillslope})(Top - Bottom)}{(X^{Hillslope} + EC50^{Hillslope})}$$

For conjugate assays the percentage of THP-1 cells forming conjugates was calculated using the formula below. The data was then fit using the same dose response equation above except with X being the percentage of THP-1 cells in conjugates, normalised where indicated.

$$Receptor\ cells\ in\ Conjugates(\%)$$
$$= \frac{Double\ positive\ events}{Receptor\ only\ events + Double\ positive\ events} * 100$$

To plot stimulation as a function of conjugation the average number of binding sites used in the stimulation assay were interpolated from the fit of the conjugation data using the four parameter dose response model. The IL-8 values from the stimulation were then plotted against the interpolated values of conjugate formation.

To enable the results from multiple experiments to be included in single plots data were normalised. This increased the statistical power of the experiments by increasing the number of data points for each fit. For statistical analysis F tests, t tests ANOVA were performed as appropriate and results presented with the following symbols: ns, not significant ; *$p < 0.05$; **$p < 0.01$; ***$p < 0.001$; ****$p < 0.0001$.

## Reporting summary

Further information on research design is available in the Nature Portfolio Reporting Summary linked to this article.

## Data availability

The source data underlying the graphs in the study can be found in Supplementary Data.

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

## Acknowledgements
We acknowledge Mark Howarth for providing Strep-Tactin, streptavidin, SpyTag and SpyCatcher constructs and for helpful discussions and advice. We thank Marion H Brown and Omer Dushek and their groups for helpful discussion. This work was supported by a Wellcome Trust Senior Investigator Award (P.A.v.d.M., grant reference101799/Z/13:/Z) and a Nuffield Medical Fellowship from the Australian Academy of Science (J.G., grant reference: #1016848).

## Author contributions
The study was conceived by P.A.v.d.M. and J.G., the experiments were performed by M.I.B. with reagents prepared by R.L.P. and E.M.D., the paper was drafted by M.I.B. and P.A.v.d.M. and finalised by all authors.

## Competing interests
PAvdM is a founder and consultant for MatchBio Limited.
