## [Peer Review File · Communications Biology]

Reviewers' comments:

Reviewer #1 (Remarks to the Author):

General Comments:

The manuscript introduces a novel system based on SpyTag/SpyCatcher technology to study non-catalytic tyrosine phosphorylated receptor (NTR or immunoreceptor) signaling of the immune system. A firm understanding of immunoreceptor triggering is critical for interrogating immune function in health and disease, yet the predominant signaling mechanism among three common models has remained highly debated: 1) kinetic segregation of phosphatases, 2) mechanotransduction, and 3) receptor clustering. The authors propose that their system can evaluate the NTR triggering mechanism independent of natural ligands, by co-incubating cells expressing either StrepTagII-fused NTRs or StrepTactin-displaying ligands. They evaluate their system on 4 NTRs from diverse families, which all signal through DAP12 adaptors and ITAM motifs (SIRP β 1, TREM-1, NKp44, Siglec-14). Their results demonstrate that increasing ligand length decreases NTR activation for these receptors, while increasing ligand mobility and valency has minimal effects. Their system addresses the need for a generalizable approach to study immunoreceptor triggering across diverse NTRs, particularly those with poorly characterized ligands and high biological importance.

However, while the authors claim support for the kinetic segregation model as the predominant mechanism, the manuscript lacks sufficient evidence in support of phosphatase exclusion. Additional experiments with and without phosphatase activity (using CD45 inhibitors, knockouts, truncations, etc.) and/or a direct quantification of phospho-Tyrosine changes under these conditions would strengthen their claims. Furthermore, the authors' selection of 4 receptors with identical downstream signaling pathways (all DAP12 adaptors) makes it difficult to generalize their results to other NTR members. It would also help reviewer confidence in the system to compare StrepTagII-StrepTactin signaling to the natural ligands, both on primary and different cell lines, as these variables could influence NTR triggering. Some NTRs (ie – NKp44) have known natural ligands or antibody binders that can be used for this comparison. The manuscript in its present form has limited biological insight and technological advancement.

Specific Comments and Suggestions:

- Phospho-signaling data would help in all experiments, with/without phosphatase activity.
- Legends in the Figures to clarify which samples are in each experiment would enhance readability of the manuscript.
- Data normalization, especially plots with both normalized IL-8 secretion and cell

conjugation, appear to exaggerate data variation and statistical significance in a way that does not match visual inspection. Raw data plots or an improved justification for their data processing methods would be helpful.

- Have the authors performed experiments directly measuring ligand expression on CHO cells with antibody-fluorophore staining, rather than biotin-488 quenching? Comparable expression levels of ligands is critical, and direct staining could reduce unpredictable factors from the biotin-488 quenching process.
- Typos and language - Page 6, Line 165 (should say “with and without stimulation”); Page 8, Line 210 (difficult to say “abrogated” activation when activation is more “attenuated”); Page 8, 239 (should say “little impact on its ability to mediate activation...”)
- sFigure 5 and Figure 3 - if each anchor ligand (CD80, CD43, CD52) have half-lives under 8 hours, were stimulation times other than 20 hours evaluated? Measuring phospho-signaling could be useful for measuring earlier timepoints.
- sFig4A, sFig6B, sFig9B - since KD is not a linear value, KD cannot be plotted with 0 at the origin and linear increments. Translating all values into log would enable more accurate comparisons.
- Figure 4 and 5 - the claim that increasing valency does not increase NTR activation is questionable, given that higher NTR activation is observed with fewer binding sites. There could be an optimal geometry for NTR activation that is not explored. For example, when StrepTactin tetramer is on the 2D cell surface, two binding sites face towards the membrane and two away. Only two binding sites might efficiently interact with the target NTR (possibly explaining why 3 of 4 NTRs activate well). While sampling every geometry is impossible, a stepwise progression through, mono-, duo-, tri-, and tetra-valent StrepTactin would provide a more thorough evaluation of valency.

Reviewer #2 (Remarks to the Author):

The manuscript “Ligand Requirements for Immunoreceptor Triggering” uses a generic Spy/SpyCatcher–Strep-Tactin linker to couple cell surface proteins on a CHO cell to immunoreceptors on THP-1 cells. THP-1 cell activation is measured by IL-8 secretion as a function of three ligand binding properties: height, mobility, and valency. The authors conclude that increasing ligand length or valency decreases NTR triggering, while changing ligand mobility does not affect triggering. The study uses a well-characterized system with clear readouts that show the effect of the ligand modifications on THP-1 cell signaling. I find some of the interpretation of results and claims about models of immunoreceptor triggering to be somewhat over simplified. This generic immunoreceptor ligand holds

promise in illuminating general mechanisms of NTR triggering, but I believe the authors dismiss some intriguing observations in favor of making clean and concise conclusions. Overall, I think the work is very nicely done and only have some minor specific points:

1. Key experimental choices are not described, which leaves readers less familiar with these systems wondering. Why are the authors studying THP-1 cells to measure signaling through NTRs? Why is IL-8 secretion an appropriate readout, and sufficient as a stand-alone measure of cellular activation? Providing these justifications will help a wider range of readers understand and app.

2. Similarly, the four receptors used are briefly described, but the description doesn't necessarily serve understanding of why comparing signaling through these receptors in the same cell type would be interesting. The text says that they are representative of four distinct NTR families. With a generic ligand, should we expect similar outcomes through these NTRs as a function of ligand properties? What are notable differences between the receptors that may complicate a simple interpretation (e.g. Siglec14 not being sensitive to ligand height but SIRP β gives very different IL-8 secretion profiles with varied ligand height)? This is addressed somewhat in the section of ligand height, comparing ligands with 3 vs. 1 extracellular domains, but is generally absent in the paper. The discussion states that this system will facilitate comparisons between NTRs. This should be demonstrated within this manuscript.

3. I agree with the authors that results from Section 1: "Varying Ligand Length" and Section 3: "Varying Ligand Valency" can be interpreted in the context of kinetic segregation and receptor aggregation models for NTR triggering, respectively. I disagree that Section 2: "Varying Ligand Mobility" (i) actually tests physiologically-relevant differences in ligand mobility and (ii) yields results that can be interpreted in relation to a model of force-dependent NTK triggering.

a. The authors compare their data to results from Gohring et al. 2021. In this paper, mobility of the supported bilayer is changed by a factor of over 1000 to change the force that the T cell receptor is experiencing ($0.72 \mu\text{m}^2/\text{s}$ to $1.3 \times 10^{-4} \mu\text{m}^2/\text{s}$). In the current manuscript, mobility is changed by about 50%, with all ligands in the $0.1\text{-}0.2 \mu\text{m}^2/\text{s}$ range. In native membranes, pMHC and TCR mobility falls in the range of $0.02\text{-}0.04 \mu\text{m}^2/\text{s}$ (Govern et al. 2010 PNAS 107:8724), with the intracellular portion of pMHC notably decreasing its mobility (Wade et al. 1989 J. Cell. Biol. 109:3325). I find it hard to believe that the range of mobility captured in this study would change the force on the receptor much compared with natural variations in physiological systems. It would be helpful if there were literature precedent supporting such sensitivity to mobility to which the authors could point.

b. Moreover, the notable difference in activation is seen between ligands of the same mobility. To me, this result points to another, unspecified attribute of the ligand resulting from the choice of anchor that causes a difference in NTR signaling. I think this is an interesting result and worth exploring.

c. It would be useful to see the FRAP data and fit used to measure mobility as an SI figure

d. In general, with this experimental system, I feel the authors do not seem to be isolating mobility as a testable parameter nearly as well as with the other parameters they study, and they do not demonstrate any connection to force. In my opinion, it would improve the paper to take a more reserved approach to conclusions on this part.

Reviewer #3 (Remarks to the Author):

In this manuscript, the authors investigated the effect of ligand length, mobility, and valency on the activation of four NTR members: SIRP β 1, Siglec 14, NKp44, and TREM-1 using a cell-surface generic ligand system. They conducted sophisticated experiments and found that increasing ligand length enhances cell-cell conjugation but impairs activation via NTRs, while varying ligand mobility or valency has minimal effect on activation.

While this study offers potential insights to advance chimeric NTR design for cellular-based immunotherapies, there are several concerns that need to be addressed:

1. It is essential to determine whether the interaction between artificial ligand and tagged receptor accurately represents native ligand-receptor interaction and downstream signaling pathway stimulation. Validating findings using native receptors interacting with cognate ligands would strengthen the study's credibility.

2. The rationale for studying the mechanistic interactions of these specific NTRs among a diverse array of NTRs is not entirely clear. The shared feature between all 4 NTRs is the signaling through adaptor DAP12, thus further characterization of DAP12 phosphorylation or other relevant readouts indicative of NTR/DAP12 signaling pathway activation would enhance its significance.

3. The use of IL8 secretion in THP1 cells (monocyte-like cell line) to demonstrate activation of all four NTRs may not fully capture the functionality of NKp44, which is specific to NK cell activation and cytotoxicity. Utilizing NK-like cell lines to assess NKp44 ligand stimulation and measure NK cell activation and cytotoxicity would provide more

physiologically-relevant data.

4. The observation that receptor clustering induced by tetravalent ligand does not impact NTR activation requires validation through imaging to confirm receptor clustering upon ligand ligation.

Responses to reviewer comments

Manuscript COMMSBIO-24-0198-T
Li et al

We thank the referees for these helpful and constructive comments, which have enabled us to improve the manuscript.

Reviewer #1 (Remarks to the Author):

General Comments:

The manuscript introduces a novel system based on SpyTag/SpyCatcher technology to study non-catalytic tyrosine phosphorylated receptor (NTR or immunoreceptor) signaling of the immune system. A firm understanding of immunoreceptor triggering is critical for interrogating immune function in health and disease, yet the predominant signaling mechanism among three common models has remained highly debated: 1) kinetic segregation of phosphatases, 2) mechanotransduction, and 3) receptor clustering. The authors propose that their system can evaluate the NTR triggering mechanism independent of natural ligands, by co-incubating cells expressing either StrepTagII-fused NTRs or StrepTactin-displaying ligands. They evaluate their system on 4 NTRs from diverse families, which all signal through DAP12 adaptors and ITAM motifs (SIRPβ1, TREM-1, NKp44, Siglec-14). Their results demonstrate that increasing ligand length decreases NTR activation for these receptors, while increasing ligand mobility and valency has minimal effects. Their system addresses the need for a generalizable approach to study immunoreceptor triggering across diverse NTRs, particularly those with poorly characterized ligands and high biological importance.

However, while the authors claim support for the kinetic segregation model as the predominant mechanism, the manuscript lacks sufficient evidence in support of phosphatase exclusion. Additional experiments with and without phosphatase activity (using CD45 inhibitors, knockouts, truncations, etc.) and/or a direct quantification of phospho-Tyrosine changes under these conditions would strengthen their claims.

We acknowledge that we have not provided support for phosphatase exclusion. This is beyond the scope this study, which was focused on examining which ligand binding properties were important for triggering. We have clarified throughout that our finding that increasing ligand length abrogated triggering is consistent with the involvement of kinetic segregation model but does not prove it. We have also noted in the Discussion that other predictions of the KS model need to be tested in future work.

Furthermore, the authors' selection of 4 receptors with identical downstream signaling pathways (all DAP12 adaptors) makes it difficult to generalize their results to other NTR members. It would also help reviewer confidence in the system to compare StrepTagII-StrepTactin signaling to the natural ligands, both on primary and different cell lines, as these variables could influence NTR triggering. Some NTRs (ie – NKp44) have known natural ligands or antibody binders that can be used for this comparison. The manuscript in its present form has limited biological insight and technological advancement.

We agree that confirming the results obtained with generic ligands with natural ligands would strengthen the conclusions. Indeed, we have previously validated the generic ligand system in this way. In Figure 4 of (Denham et al., 2019) we compared quantitatively the activation of a TCR

by our generic ligand and its native pMHC ligand. The surface density of generic ligand and native ligand required for activation were indistinguishable, indicating that generic ligand was equivalent to the native ligand. Unfortunately, it is more challenging to do such validation for other NTRs, because they can easily be titrated. We have recently developed a system that should allow this, but this would be an extensive amount of work, beyond the scope of the current study. These points are made in the Introduction (lines 78-80) and in the final Discussion paragraph.

Specific Comments and Suggestions:

- Phospho-signaling data would help in all experiments, with/without phosphatase activity.

While these would be interesting, these experiments are extremely challenging to do in the high throughput manner necessary for our titration experiments.

- Legends in the Figures to clarify which samples are in each experiment would enhance readability of the manuscript.

We have added these to all Figures

- Data normalization, especially plots with both normalized IL-8 secretion and cell conjugation, appear to exaggerate data variation and statistical significance in a way that does not match visual inspection. Raw data plots or an improved justification for their data processing methods would be helpful.

We have now explained in the Data Analysis section that data were normalised to enable the results from multiple experiments to be included in single plots. By increasing the number of data points for each fit, this increased the statistical power of the experiments.

- Have the authors performed experiments directly measuring ligand expression on CHO cells with antibody-fluorophore staining, rather than biotin-488 quenching? Comparable expression levels of ligands is critical, and direct staining could reduce unpredictable factors from the biotin-488 quenching process.

We did not use antibody-fluorophore staining as we lacked a suitable antibody to either SpyCatcher or Streptactin. Note however that biotin-488 quenching was used to measure the Kd of coupling for each ligand. Any differences in quenching between ligands would not be expected to have any impact on this Kd value. Binding site number depends on our estimate of the maximum number of binding sites using the Biotin-4-fluorescein titration method. Because to the nature of these titration experiments, differences in quenching would not be expected to have an impact on these estimates.

- Typos and language - Page 6, Line 165 (should say “with and without stimulation”); Page 8, Line 210 (difficult to say “abrogated” activation when activation is more “attenuated”); Page 8, 239 (should say “little impact on its ability to mediate activation...”)

We have corrected these

- sFigure 5 and Figure 3 - if each anchor ligand (CD80, CD43, CD52) have half-lives under 8 hours, were stimulation times other than 20 hours evaluated? Measuring phospho-signaling could be useful for measuring earlier timepoints.

We have not done shorter stimulations. Given that CD52 is more potent than CD80 as a ligand anchor, despite having a shorter half-life, it is unlikely that these will be informative.

- sFig4A, sFig6B, sFig9B - since KD is not a linear value, KD cannot be plotted with 0 at the origin and linear increments. Translating all values into log would enable more accurate comparisons.

We have converted these to log scales.

- Figure 4 and 5 - the claim that increasing valency does not increase NTR activation is questionable, given that higher NTR activation is observed with fewer binding sites. There could be an optimal geometry for NTR activation that is not explored. For example, when StrepTactin

tetramer is on the 2D cell surface, two binding sites face towards the membrane and two away. Only two binding sites might efficiently interact with the target NTR (possibly explaining why 3 of 4 NTRs activate well). While sampling every geometry is impossible, a stepwise progression through, mono-, duo-, tri-, and tetra-valent StrepTactin would provide a more thorough evaluation of valency.

We do not mean to claim that increasing valency does not increase NTR activation. Indeed, we show that it increasing valency does increase NTR activation. Importantly, we also show that increasing valency increases NTR binding, as measured by cell-cell conjugation. This is as expected if there is multivalent binding, which increases avidity. We then show that, once one controls for the increase in NTR binding that results from increasing valency, increasing valency does not independently increase in IL-8 release. We have clarified this by making suitable revisions.

Reviewer #2 (Remarks to the Author):

The manuscript “Ligand Requirements for Immunoreceptor Triggering” uses a generic Spy/SpyCatcher–Strep-Tactin linker to couple cell surface proteins on a CHO cell to immunoreceptors on THP-1 cells. THP-1 cell activation is measured by IL-8 secretion as a function of three ligand binding properties: height, mobility, and valency. The authors conclude that increasing ligand length or valency decreases NTR triggering, while changing ligand mobility does not affect triggering. The study uses a well-characterized system with clear readouts that show the effect of the ligand modifications on THP-1 cell signaling. I find some of the interpretation of results and claims about models of immunoreceptor triggering to be somewhat over simplified. This generic immunoreceptor ligand holds promise in illuminating general mechanisms of NTR triggering, but I believe the authors dismiss some intriguing observations in favor of making clean and concise conclusions. Overall, I think the work is very nicely done and only have some minor specific points:

1. Key experimental choices are not described, which leaves readers less familiar with these systems wondering. Why are the authors studying THP-1 cells to measure signaling through NTRs? Why is IL-8 secretion an appropriate readout, and sufficient as a stand-alone measure of cellular activation? Providing these justifications will help a wider range of readers understand and app.

We have added a justification for using THP-1 cells and IL-8 release, together with references, to appropriate parts of the Results section.

2. Similarly, the four receptors used are briefly described, but the description doesn't necessarily serve understanding of why comparing singling through these receptors in the same cell type would be interesting. The text says that they are representative of four distinct NTR families. With a generic ligand, should we expect similar outcomes through these NTRs as a function of ligand properties? What are notable differences between the receptors that may complicate a simple interpretation (e.g. Siglec14 not being sensitive to ligand height but SIRP β gives very different IL-8 secretion profiles with varied ligand height)? This is addressed somewhat in the section of ligand height, comparing ligands with 3 vs. 1 extracellular domains, but is generally absent in the paper. The discussion states that this system will facilitate comparisons between NTRs. This should be demonstrated within this manuscript.

All these receptors and other members of the family that they represent can be expressed on monocytes, and THP-1 cells are a human monocyte cell line. One focus of the study is to compare the ligand requirement for these different receptors. To ensure that the comparison is

well controlled we express them in the same cell line, stimulate them with the same ligands, and use the same readout. We also focused on receptors that use the same DAP-12 signalling subunit.

This enabled us to identify subtle differences in Siglec-14 and the other NTRs, as well as differences between the short and long receptors and relate them to differences in valency (Siglec-14 is a disulphide linked homodimer).

These findings show that generic ligand system enables us to compare NTR to the same ligand and identify subtle differences. We have previously used the generic ligand system to show that the TCR is inherently more sensitive than other NTRs (Denham et al., 2019).

The shared properties in response to the same generic ligand are not unexpected as these NTRs deploy the same signalling machinery (associated DAP12).

We have clarified these points in the Introduction and Discussion.

3. I agree with the authors that results from Section 1: “Varying Ligand Length” and Section 3: “Varying Ligand Valency” can be interpreted in the context of kinetic segregation and receptor aggregation models for NTR triggering, respectively. I disagree that Section 2: “Varying Ligand Mobility” (i) actually tests physiologically-relevant differences in ligand mobility and (ii) yields results that can be interpreted in relation to a model of force-dependent NTK triggering.

a. The authors compare their data to results from Gohring et al. 2021. In this paper, mobility of the supported bilayer is changed by a factor of over 1000 to change the force that the T cell receptor is experiencing ($0.72 \mu\text{m}^2/\text{s}$ to $1.3 \times 10^{-4} \mu\text{m}^2/\text{s}$). In the current manuscript, mobility is changed by about 50%, with all ligands in the 0.1 - $0.2 \mu\text{m}^2/\text{s}$ range. In native membranes, pMHC and TCR mobility falls in the range of 0.02 - $0.04 \mu\text{m}^2/\text{s}$ (Govern et al. 2010 PNAS 107:8724), with the intracellular portion of pMHC notably decreasing its mobility (Wade et al. 1989 J. Cell. Biol. 109:3325). I find it hard to believe that the range of mobility captured in this study would change the force on the receptor much compared with natural variations in physiological systems. It would be helpful if there were literature precedent supporting such sensitivity to mobility to which the authors could point.

This point is well made. We acknowledge this in the text and point out that even lipid anchored proteins diffuse more slowly in cells than in model membrane, likely because membrane proteins immobilized by attachment to the cortical actin cytoskeleton form obstructions, which have been termed ‘picket fences’. We have revised the Discussion to make these points.

b. Moreover, the notable difference in activation is seen between ligands of the same mobility. To me, this result points to another, unspecified attribute of the ligand resulting from the choice of anchor that causes a difference in NTR signaling. I think this is an interesting result and worth exploring.

We acknowledge this and have modified the Discussion accordingly.

c. It would be useful to see the FRAP data and fit used to measure mobility as an SI figure

We have added this data (new sFig. 4A)

d. In general, with this experimental system, I feel the authors do not seem to be isolating mobility as a testable parameter nearly as well as with the other parameters they study, and they do not demonstrate any connection to force. In my opinion, it would improve the paper to take a more reserved approach to conclusions on this part.

We agree that these conclusions should be more tentative, and have revised the text

accordingly throughout, especially in the relevant section of the Discussion.

Reviewer #3 (Remarks to the Author):

In this manuscript, the authors investigated the effect of ligand length, mobility, and valency on the activation of four NTR members: SIRP β 1, Siglec 14, NKp44, and TREM-1 using a cell-surface generic ligand system. They conducted sophisticated experiments and found that increasing ligand length enhances cell-cell conjugation but impairs activation via NTRs, while varying ligand mobility or valency has minimal effect on activation.

While this study offers potential insights to advance chimeric NTR design for cellular-based immunotherapies, there are several concerns that need to be addressed:

1. It is essential to determine whether the interaction between artificial ligand and tagged receptor accurately represents native ligand-receptor interaction and downstream signaling pathway stimulation. Validating findings using native receptors interacting with cognate ligands would strengthen the study's credibility.

Reviewer 1 made a similar point to which we have responded. For convenience we have copied this response below.

We agree that confirming the results obtained with generic ligands with natural ligands would strengthen the conclusions. Indeed, we have previously validated the generic ligand system in this way. In Figure 4 of (Denham et al., 2019) we compared quantitatively the activation of a TCR by our generic ligand and its native pMHC ligand. The surface density of generic ligand and native ligand required for activation were indistinguishable, indicating that generic ligand was equivalent to the native ligand. Unfortunately, it is more challenging to do such validation for other NTRs, because they can easily be titrated. We have recently developed a system that should allow this, but this would be an extensive amount of work, beyond the scope of the current study. These points are made in the Introduction (lines 78-80) and in the final Discussion paragraph.

2. The rationale for studying the mechanistic interactions of these specific NTRs among a diverse array of NTRs is not entirely clear. The shared feature between all 4 NTRs is the signaling through adaptor DAP12, thus further characterization of DAP12 phosphorylation or other relevant readouts indicative of NTR/DAP12 signaling pathway activation would enhance its significance.

The aim of the current study was to examine the impact of varying the ligand length, valency and mobility on NTR triggering. NTRs have structurally diverse ectodomains and ligands and associate with a variety of signalling subunits. We chose representative NTRs from 4 structurally different families of NTR, to see whether they shared the same dependence on these features. We chose NTRs that use the same DAP-12 signalling subunits to simplify interpretation of the results. Since all the NTRs chosen shared the same signalling subunits and were stimulated by identical ligands, any differences observed could more easily be ascribed to differences between the NTRs (such as size or being a dimer). The focus of this study was the ligand requirements of NTR triggering, which required high throughput assays to enable ligand titrations to be performed. While interesting, more detailed studies of the DAP-12 signalling

pathways unfortunately cannot be performed in a high throughput manner, and so are beyond the scope of the current study.

3. The use of IL8 secretion in THP1 cells (monocyte-like cell line) to demonstrate activation of all four NTRs may not fully capture the functionality of NKp44, which is specific to NK cell activation and cytotoxicity. Utilizing NK-like cell lines to assess NKp44 ligand stimulation and measure NK cell activation and cytotoxicity would provide more physiologically-relevant data. We agree that it would be interesting to explore NKp44 in NK cells. However, a central purpose of this paper was to compare these NTRs with each other under controlled conditions, and this required expressing them in the same cell type.

4. The observation that receptor clustering induced by tetravalent ligand does not impact NTR activation requires validation through imaging to confirm receptor clustering upon ligand ligation.

Any receptor/ligand interaction at a cell-cell interface where both molecules are mobile will induce clustering. Unfortunately detecting very close (<10 nm) clustering of the sort induced just by multivalent ligands is extremely challenging, especially at a cell-cell interface. We have considered and explored imaging and concluded that this would require a program of work using a modified system, such as planar supported lipid bilayers, in collaboration with experts in advanced imaging modalities, and is thus beyond the scope of this work.

We would argue that the fact that the tetravalent ligand is so much more effective at inducing cell-cell conjugation than the monovalent ligand is compelling evidence that it is binding multivalently to NTRs, and thus inducing physical clustering of two or more NTRs.

REFERENCES

Denham EM, Barton MI, Black SM, Bridge MJ, Wet B de, Paterson RL, Merwe PA van der, Goyette J. 2019. A generic cell surface ligand system for studying cell–cell recognition. *Plos Biol* **17**:e3000549. doi:10.1371/journal.pbio.3000549

Reviewers' comments:

Reviewer #1 (Remarks to the Author):

The authors have clarified several points in the manuscript and it is acceptable for publication.

Reviewer #2 (Remarks to the Author):

My concerns have been sufficiently addressed

Reviewer #3 (Remarks to the Author):

In the abstract, the authors showed that "NTR signaling requires phosphorylation of cytoplasmic tyrosine residues by SRC-family tyrosine kinases. How ligand binding to NTRs induces this phosphorylation, also called NTR triggering, remains controversial." However, the authors did not provide evidence of NTR triggering, such as Src-mediated phosphorylation, which is an essential and direct measure of NTR triggering as stated in their abstract. Therefore, the key conclusion in this paper lacks sufficient evidence.

While I understand the authors' point that measuring phosphorylation as a quantitative readout is challenging compared to IL8 secretion, relying solely on IL-8 as an NTR triggering indicator is concerning. It's important to characterize other pro-inflammatory cytokines to strengthen the conclusion of NTR triggering.

Furthermore, the authors overexpressed both engineered NTR and DAP12 in the THP1 cell line to measure IL-8, indicating that native DAP12 signaling may not be induced under this experimental setup and requires DAP12 overexpression. This raises concerns about whether the kinetics-segregation model proposed from such settings can be applied to physiological NTR triggering with naïve receptors, ligands, and DAP12.

Responses to reviewer comments

Manuscript COMMSBIO-24-0198-T

Li et al

Reviewer #1 (Remarks to the Author):

The authors have clarified several points in the manuscript and it is acceptable for publication.

Reviewer #2 (Remarks to the Author):

My concerns have been sufficiently addressed

We thank these reviewers for accepting our revisions

Reviewer #3 (Remarks to the Author):

In the abstract, the authors showed that "NTR signaling requires phosphorylation of cytoplasmic tyrosine residues by SRC-family tyrosine kinases. How ligand binding to NTRs induces this phosphorylation, also called NTR triggering, remains controversial." However, the authors did not provide evidence of NTR triggering, such as Src-mediated phosphorylation, which is an essential and direct measure of NTR triggering as stated in their abstract. Therefore, the key conclusion in this paper lacks sufficient evidence.

While it would indeed be of interest to directly examine SRC-family tyrosine kinase mediated phosphorylation of these NTRs, such experiments are unfortunately not possible to perform in the high-throughput manner required for our titration-based analysis. Downstream readouts (such as IL-8 release) are widely accepted for examining cell-surface receptor activation by ligands. We introduce the concept of NTR triggering mechanisms in the abstract to justify our analysis of the effect of ligand size, mobility and valency on activation by NTRs. We suggest that there is no plausible alternative mechanism by which changing these properties could impact a downstream response such as IL-8 release.

While I understand the authors' point that measuring phosphorylation as a quantitative readout is challenging compared to IL8 secretion, relying solely on IL-8 as an NTR triggering indicator is concerning. It's important to characterize other pro-inflammatory cytokines to strengthen the conclusion of NTR triggering.

While analysis of multiple downstream responses is often of interest in cell stimulation assays, it is not obvious why this would provide any useful information for the purposes of the current study, which is to examine the impact of ligand properties on NTR signal transduction.

Furthermore, the authors overexpressed both engineered NTR and DAP12 in the THP1 cell line to measure IL-8, indicating that native DAP12 signaling may not be induced under this experimental setup and requires DAP12 overexpression. This raises concerns about whether the kinetics-segregation model proposed from such settings can be applied to physiological NTR triggering with naïve receptors, ligands, and DAP12.

This is an interesting point. We note that NTRs with charged transmembrane domains, such as the ones expressed in this study, require association with signalling subunit dimers with appropriate complementary charges for expression (Call and Wucherpfennig, 2007).

Because expression of endogenous DAP-12 is likely to match expression of endogenous NTRs in THP1 cells, we reasoned that it would likely be limiting and that it would be appropriate to express additional DAP-12 when expressing an additional NTR.

We acknowledge that this system is not perfectly physiological, we suggest that it is a reasonable first step for exploring ligand requirements for NTR signal transduction.

REFERENCES

Call ME, Wucherpfennig KW. 2007. Common themes in the assembly and architecture of activating immune receptors. *Nature Reviews Immunology* **7**:841–850. doi:10.1038/nri2186

Denham EM, Barton MI, Black SM, Bridge MJ, Wet B de, Paterson RL, Merwe PA van der, Goyette J. 2019. A generic cell surface ligand system for studying cell–cell recognition. *Plos Biol* **17**:e3000549. doi:10.1371/journal.pbio.3000549